# Biological invasions alter environmental microbiomes: A meta-analysis

**Antonino Malacrinò**[1]*, **Victoria A. Sadowski**[1], **Tvisha K. Martin**[2], **Nathalia Cavichiolli de Oliveira**[3], **Ian J. Brackett**[1], **James D. Feller**[1], **Kristian J. Harris**[1], **Orlando Combita Heredia**[1], **Rosa Vescio**[1,4], **Alison E. Bennett**[1]

**1** Department of Evolution, Ecology and Organismal Biology, The Ohio State University, Columbus, OH, United States of America, **2** College of Food, Agricultural and Environmental Sciences, The Ohio State University, Columbus, OH, United States of America, **3** Department of Entomology, University of Sao Paulo, Piracicaba (SP), Brazil, **4** Dipartimento AGRARIA, Università degli Studi Mediterranea, Reggio Calabria, Italy

* malacrino.1@osu.edu, antonino.malacrino@gmail.com

## Abstract

Biological invasions impact both agricultural and natural systems. The damage can be quantified in terms of both economic loss and reduction of biodiversity. Although the literature is quite rich about the impact of invasive species on plant and animal communities, their impact on environmental microbiomes is underexplored. Here, we re-analyze publicly available data using a common framework to create a global synthesis of the effects of biological invasions on environmental microbial communities. Our findings suggest that non-native species are responsible for the loss of microbial diversity and shifts in the structure of microbial populations. Therefore, the impact of biological invasions on native ecosystems might be more pervasive than previously thought, influencing both macro- and micro-biomes. We also identified gaps in the literature which encourage research on a wider variety of environments and invaders, and the influence of invaders across seasons and geographical ranges.

**Data Availability Statement:** Raw data from single studies is publicly available. The code used to perform our analyses has been deposited on

## Introduction

Biological invasions have severe impacts on biodiversity, community composition and ecosystem functions [1–5]. Invasive plants can alter many important ecosystem functions including the nitrogen cycle [6], carbon cycle, and decomposition. For example, invasion by the plant Amur Honeysuckle altered the decomposition rate in the invaded environment likely through changes in litter quality [7]. Exotic snails have been found to alter carbon and nitrogen fluxes in freshwater systems through their consumption/excretion activity [8]. These functions are provided by environmental microbiomes. Yet, despite the implications for ecosystem functioning, we are still learning the consequences of biological invasions on environmental microbiomes.

Previous studies have shown biological invasions can impact the diversity and taxonomical structure of environmental microbiomes. For example, we often see a shift in soil microbiota following invasion by non-native plant species [9–19]. Removal of feral pigs increased the

GitHub: https://github.com/amalacrino/Biol_
Invasion_Microb_MA

**Funding:** The author(s) received no specific
funding for this work.

**Competing interests:** The authors have declared
that no competing interests exist.

diversity of soil bacterial communities and shifted their structure [20], and invasive crusta-
ceans [21], mussels [22] and jellyfish [23] produce changes in the structure of water micro-
biomes. However, shifts in environmental microbiome as consequence of biological invasions
do not always occur. For example, invasion by the plants *Robinia pseudoacacia* [24], *Eucalyptus*
sp. [25], and *Vincetoxicum rossicum* [26] did not alter the structure of soil microbial communi-
ties. Also, some microcosms exposed to the simultaneous invasion of multiple plant species
[27–29] did not alter soil microbiomes. Similarly, soil microbiome structure in microcosms
did not change with the addition of the invasive earthworm *Aporrectodea trapezoides* [30]. Sev-
eral of these studies used techniques (e.g. DGGE, PLFA, t-RFLP) that limit fine scale investiga-
tions of biological invasions on environmental microbiome diversity and taxonomical
composition. Among the studies using high-throughput amplicon-sequencing techniques,
most did not find changes in microbiome diversity [13, 15, 18, 19, 21, 25, 27, 28, 30], few
reported a decrease of microbial diversity in response to invasion [16, 20, 29], and fewer still
reported an increase [11, 14]. Thus, there is little consensus on the effects of biological inva-
sions on the diversity and taxonomical structure of the environmental microbiomes, both tied
to the stability and function of microbial communities [31, 32].

Our ability to draw broad conclusions from published studies is limited, because individual
studies have occurred within a limited geographical range or with a limited group of species.
Meta-analyses of published biological means have long enabled more robust conclusions than
individual studies [33–36]. However, the meta-analytic approach has less frequently been
applied to amplicon-sequencing data that represent environmental microbiome community
composition. The majority of meta-analytic metabarcoding studies have occurred in the medi-
cal sciences [37–45]. This approach can be successfully used to address ecological questions.
For example, meta-analytic metabarcoding studies have found common patterns in the struc-
ture of indoor microbiomes [46] and freshwater eukaryotes [47]. Shade et al. [48] also used a
meta-analysis of metabarcoding datasets from different environments highlighting a time-
dependent structure of microbiomes. A meta-analytic approach has also been used to test the
effects of stressors (e.g. water availability, temperature, heavy metals) on environmental micro-
biomes [49]. Thus meta-analyses on microbiome data have a striking potential to address
global-scale questions, generate new hypotheses and model common patterns [50], because
they provide across study comparisons [39, 51, 52].

Here, we aim to test whether the effect of biological invasion on environmental micro-
biomes can be generalized or is idiosyncratic. To do so, we collected publicly available data
and re-analyzed this data under a common framework. We tested the effect of invasive species
on the diversity and structure of environmental microbiomes, with the hypothesis that the
presence of invasive species will decrease microbial diversity and alter the composition of the
environmental microbiome. We then investigated whether certain taxonomical groups are
more responsive to biological invasions.

## Methods

### Data collection

We searched for metabarcoding studies that evaluated the effect of biological invasions on
environmental microbiomes, and compared invaded and non-invaded habitats. Our literature
search for this study was conducted using Web of Science Core Collection (accessed on March
6th, 2020) using the keywords "Invasive speci*" and "microbio*" published between 2010–
2020, and found 1,471 studies. Two additional studies were added by searching the same key-
words on Google Scholar (S1 Fig). Records were manually filtered based on the study design
appropriate for our research question. This step yielded 22 studies, and we further filtered

these studies based on data availability in public repositories. When data was not available, we attempted to contact the corresponding author. Finally we selected only studies that used the 16S rRNA marker gene, primer pair 515F/806R [53] or 341F/785R [54], and Illumina MiSeq sequencing platform. After discarding studies that failed quality checks (see below), we were able to include a total of five studies (Table 1), summing up to a total of 356 samples. The study by Gibbons et al. [28] tested the impact of five invasive plant species (*Agropyron cristatum*, *Bromus tectorum*, *Sisymbrium altissimum*, *Erodium cicutarium* and *Poa bulbosa*) on soil microbiome using microcosms, comparing monocultures of each one of them towards a mixture of eight native plant species. A similar question was tested in Rodrigues et al. [19] in field condition. They identified three locations invaded by three different exotic plant species (*Microstegium vimineum*, *Rhamnus davurica* and *Ailanthus altissima*) and, within each location, they sampled soil form an invaded area and a non-invaded area for comparisons. Similarly, Collins et al. [11] compared the soil microbial community of field sites invaded by *Artemisia rothrockii* to non-invaded sites. The study by Wehr et al. [20] focused on the effects of feral pig (*Sus scrofa*) invasion on soil microbiome comparing invaded areas to those where pigs were removed over a ~25 year chronosequence. Finally, the only study performed in an aquatic environment [22] compared water samples collected in lake areas invaded by the exotic mussel *Dreissena bugensis* to non-invaded sites. Three studies focused on invasive plants, and the remaining studies focused on a mammal and a mussel (Table 1).

We took the following steps to alleviate some of the potential sources of bias due to studies performed in different labs, using different protocols and sequenced on different instruments. First, all studies included were performed using the Illumina MiSeq platform, in order to reduce the potential bias that might be generated by directly comparing data obtained from different platforms. Second, all studies targeted the same region of the 16S rRNA, as several primer pairs targeting different regions are currently published and widely used. Three out of five papers we considered in our analysis used the 515F/806R primer pair [53], while two used the 341F/785R [54]. Because these primer pairs overlap in the V4 region of 16S rRNA we feel confident that the chance of including spurious OTUs in our analysis is quite negligible. To account for study-specific variances due to small differences in sampling procedures and lab protocols, we also included the study itself, the environment where the study was performed (i.e., soil or water) and the identity of the invasive species as stratification variables in the PERMANOVA and as random factors in our linear model. This allowed us to ensure that our results are not biased by study-specific features.

Once the papers were selected, we assigned each a "Study ID" and collected meta-data from each sample in each study (invasive species, type of organism, invaded environment). We then downloaded data from repositories using SRA Toolkit 2.10.4 for data on the SRA databases, or by manually downloading files from the MG-RAST database.

**Table 1. Summary of studies included in the meta-analysis.**

| Study ID | Invasive organism | Species | Invaded environment | Reference |
|---|---|---|---|---|
| MPG13011 | Plant | *Agropyron cristatum*, *Bromus tectorum*, *Sisymbrium altissimum*, *Erodium cicutarium* and *Poa bulbosa* | Soil | Gibbons et al. [28] |
| MPG87547 | Mammal | *Sus scrofa* | Soil | Wehr et al. [20] |
| PRJNA296487 | Plant | *Microstegium vimineum*, *Rhamnus davurica* and *Ailanthus altissima* | Soil | Rodrigues et al. [19] |
| PRJNA320310 | Plant | *Artemisia rothrockii* | Soil | Collins et al. [11] |
| PRJNA385848 | Mussel | *Dreissena bugensis* | Water | Denef et al. [22] |

### Data processing and analysis

Paired-end reads were merged using FLASH 1.2.11 [55] and data were processed using QIIME 1.9.1 [56]. Quality-filtering of reads was performed using default parameters, binning OTUs and discarding chimeric sequences identified with VSEARCH 2.14.2 [57]. Taxonomy for representative sequences was determined by querying against the SILVA database v132 [58] using the BLAST method. A phylogeny was obtained by aligning representative sequences using MAFFT v7.464 [59] and reconstructing a phylogenetic tree using FastTree [60].

Data analysis was performed using R statistical software 3.5 [61] with the packages phyloseq [62] and vegan [63]. Read counts were normalized using DESeq2 v1.22.2 [64] prior to data analysis. Singletons and sequences classified as chloroplast were excluded, as well as samples which had less than 5000 sequence counts. Shannon diversity was fit to a linear mixed-effects model specifying *sample type* (invaded or control), *organism* (plant, mammal, mussel), and their interactions as fixed factors. We included *studyID* or both *studyID* and *environment* (soil or water) as random factors, and both models reported similar results (S2 Table). We focused on the one with only *studyID* as random effect due to the lower AIC value. Models were fit using the *lmer*() function under the *lme4* package [65] and the package *emmeans* was used to infer pairwise contrasts (corrected using False Discovery Rate, FDR). Furthermore, we explored the effects of *sample type* and *organism* on the structure of the microbial communities using a multivariate approach. Distances between pairs of samples, in terms of community composition, were calculated using a Unifrac matrix, and then visualized using an RDA procedure. Differences between sample groups were inferred through PERMANOVA multivariate analysis (999 permutations). We ran two different PERMANOVA models: in one we stratified permutations at level of *studyID* and *identity of invasive species*, and in the other we stratified permutations at level of *studyID*, *environment* and *identity of invasive species*, obtaining similar results (S3 Table). Pairwise contrasts from PERMANOVA were subjected to FDR correction. Finally, the relative abundance of each bacterial family was fit using the *lmer*() function to test the effects of *sample type* (invaded or control) on individual taxa. We ran two different linear mixed-effects models, one including *studyID*, *organism* (plant, mammal, mussel) and *environment* as random factors, and another with *studyID* and *organism* as random factors, obtaining similar results (S4 Table).

## Results

Our search yielded 5 studies with an appropriate experimental design and available data, for a total of 356 samples. A few samples failed quality checks and we further considered 335 samples for downstream analyses. Sequences clustered into 22831 OTUs, after quality checks, removal of singletons and "chloroplast" reads, with an average of 61776.92 reads per sample. This high OTU count is likely a result of increased richness from analyzing samples across multiple environments (soil and water) and from different geographical regions.

Biological invasions led to a reduction in Shannon diversity (Control = 6.92±0.06, Invaded = 6.73±0.07, $\chi^2$ = 3.85, df = 1, $P$ = 0.04). We also found biological invasions altered microbiome community composition in the invaded environment compared to the control (Table 2 and Fig 1). The type of invasive organism (plant, mammal, or mussel) produced a different community structure (pairwise $P<0.01$, FDR corrected). A deeper analysis of bacterial families (S5 Table) revealed that some taxonomic groups are significantly more abundant in invaded environments (Blastocatellaceae, Chitinophagaceae, Nitrosomonadaceae, Pirellulaceae, Sphingomonadaceae), while others are more abundant in non-invaded samples (Acetobacteraceae, Beijerinckiaceae, Gemmataceae, Micromonosporaceae, Pedosphaeraceae, Solibacteraceae, Solirubrobacteraceae).

**Table 2. Results from PERMANOVA analysis testing the effects of *sample type* (invaded/control), *organism group* (plant, mammal, mussel) and their interaction on microbial community composition.** The factors *studyID* (unique for each study) was used as strata to constrain permutations.

| Factor | df | $R^2$ | F | P |
|---|---|---|---|---|
| Sample type (Invaded/Control) | 1 | 0.011 | 6.68 | <0.001 |
| Organism group (plant, mammal, mussel) | 3 | 0.411 | 118.86 | <0.001 |
| Sample type × Organism group | 3 | 0.007 | 2.1 | 0.01 |

## Discussion

Here we show biological invasions decrease the diversity of environmental microbiomes. While several studies have investigated the effects of species invasions on environmental microbiomes, we still lack a generalized consensus across different environmental microbiomes and systems. Previous studies have found that invasive species increased environmental microbial diversity [11, 14], while others reported a decrease [16, 20, 29]. However, the majority of studies did not analyze the microbial diversity, as they used techniques that did not allow for such analysis, or reported no changes [9, 10, 12, 13, 15, 17–19, 21–28, 30]. Within the studies included in our analysis, invasion by feral pigs decreased soil microbial diversity, while invasion by *Artemisia rothrockii* increased soil microbial diversity. The remaining three studies in our analysis reported no effects of biological invasions on environmental microbiome diversity. Our analysis was constrained in terms of sampled environment (soil) and invasive organism (plants), and an expanded dataset would be beneficial to generalize our results. Microbial diversity is tied to the function of microbiomes, and changes in diversity can reflect changes in function [66–68]. Changes in microbial diversity and function do not always have the same direction [69], which might explain the discrepancy between our results and other studies.

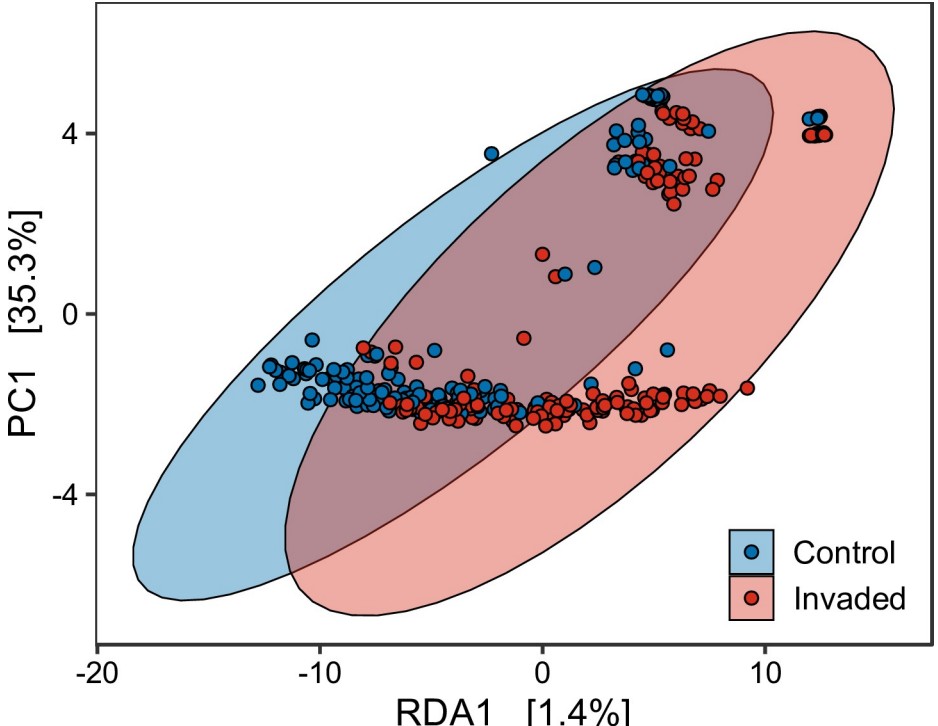

**Fig 1. RDA ordination using a Bray-Curtis distance matrix of samples.**

While we observed a relatively small reduction of the Shannon diversity index in invaded environments compared to non-invaded environment, this matches with the changes we observed in terms of community composition.

Indeed, our report of changes in community composition was relatively consistent with the published literature and the individual results of the studies we analyzed. Most studies of the influence of biological invasions on environmental microbiomes found that biological invasions alter environmental microbial community composition. However, some previous reports did not report changes [24–30], including the study by Gibbons et al. [28] considered in our analysis. This variation may be due to individual effects of organisms on the environment. For example, invasive plants may alter soil microbiome composition through root exudates [5], and invasive mussels may alter water microbiome composition via bacterial removal through their feeding activity [22]. Thus, reported influences on community composition are more consistent. Alternatively, changes in community composition might be due to the response of some bacterial groups to environmental disturbance. The bacterial families that we found to be differentially abundant between the invaded and control environments have diverse ecological functions ranging from nitrogen fixation and carbohydrate metabolism to antimicrobial properties. Although the differences in relative abundance we found might be relatively small, they can have an important impact on the functions of the environmental microbiome [70]. Many of the families that showed a significant difference in abundance between invaded and control environments include taxa that play important roles at various points during nitrogen and carbon cycling (i.e. Nitrosomonadaceae, Acetobacteraceae, Chitinophagaceae, Micromonosporaceae, Gemmataceae, Beijerinckiaceae, Pirellulaceae) [71–81]. However, nitrogen-fixing and carbohydrate-degrading bacteria did not have a unified response to invaded environments as some increased and others decreased in abundance in invaded environments. Many nitrogen-fixing bacteria have been shown to respond to environmental disturbance, such as Acidobacteria abundances during forest to pasture conversions or Pirellulaceae's response to the presence of microplastics [82, 83]. Thus, changes in environmental microbiome community composition appear to be linked to changes in ecosystem functions, although this pattern is not yet predictable across functions and taxa.

Few previous studies on biological invasions have reported details on the differential abundance of taxa, and among these we found limited general consensus. For example, some studies report a decrease in abundance of bacteria associated with nitrogen cycling (e.g. Nitrosphaeria, Nitrospira, Nitrosomonadales) [13, 14, 19], while others report an increase of Nitrosomonadaceae following invasion [30]. In our study some groups associated with the nitrogen cycle were positively associated with biological invasions (i.e. Nitrosomonadaceae, Pirellulaceae, Chitinophagaceae) while others were negatively associated (Beijerinckiaceae, Micromonosporaceae). Unfortunately, amplicon-based sequencing has a limited power to infer changes in the functions of microbiomes. Future metagenomic and metatranscriptomic studies are needed to investigate whether biological invasions alter gene content or gene expression of environmental microbiomes, and whether this reflects changes in biogeochemical cycling.

Meta-analyses are also useful to highlight gaps in the literature, and here we highlight some aspects that warrant further investigation. First, we identified a large gap in the availability of sequencing data from multiple types of environments and types of invasive species. For our analysis almost all available data came from two environments: four sets of data came from soil and one came from freshwater. Greater effort is needed for sample collection from invasions in both freshwater and marine environments. Without sufficient diversity of sample environments, it is impossible to tell whether microbial shifts following an invasion are unique to an invaded environment. Second, in our analysis the majority of data came from one type of invasive species: plants. Noticeably absent from our dataset were invasions by insects, fish, and

amphibians. Sequencing data is needed from a larger number of invasive species to allow us to broadly assess shifts in microbial community structure. A third gap we identified was the lack of spatial and temporal resolution. Almost all of the initially identified 22 studies we assessed were also restricted to one season of sampling and were conducted in the Northern Hemisphere. Thus, it is infeasible with existing datasets to validate the influence of latitude or explore how seasonality and biological invasions interact to modulate microbial communities. Thus, there are a number of opportunities for future research on how biological invasions alter environmental microbial communities.

Here we analyzed 16S amplicon sequencing data from five studies and show that biological invasions influence both the diversity and the structure of environmental microbiomes. Understanding the impact of biological invasions on environmental microbiomes is of high priority to preserve ecosystem functions [84]. We identified a number of gaps in our knowledge, including the need to assess a wider range of environments, invasive species, temporal variation, and latitudinal variation. We also demonstrate the power of re-analysis of publicly available datasets using a common pipeline which benefited from open-data initiatives.

## Supporting information

**S1 Fig. PRISMA workflow.** From: Moher D, Liberati A, Tetzlaff J, Altman DG, The PRISMA Group (2009). Preferred Reporting Items for Systematic Reviews and Meta-Analyses: The PRISMA Statement. PLoS Med 6(7): e1000097. doi:10.1371/journal.pmed1000097.
(PDF)

**S1 Table. PRIMA checklist.** From: Moher D, Liberati A, Tetzlaff J, Altman DG, The PRISMA Group (2009). Preferred Reporting Items for Systematic Reviews and Meta-Analyses: The PRISMA Statement. PLoS Med 6(7): e1000097. doi:10.1371/journal.pmed1000097.
(PDF)

**S2 Table. Comparison of two different linear mixed-effects models testing the effect of biological *sample type* (invaded or control), *organism* (plant, mammal, mussel), and their interactions, on Shannon diversity index of the environmental microbiome.** In Model 1 we included *studyID* as random factor, while in Model 2 we included both *studyID* and *environment* (soil or water) as random factors.
(PDF)

**S3 Table. Comparison of two different PERMANOVA models testing the effect of biological *sample type* (invaded or control), *organism* (plant, mammal, mussel), and their interactions, on the structure of the environmental microbiome.** In Model 1 we included *studyID* and *speciesID* as stratification factor, while in Model 2 we included *studyID*, *speciesID* and *environment* (soil or water) as stratification factors.
(PDF)

**S4 Table. Results from two different the linear mixed-effects model testing the abundance of each bacterial family against *sample type* (invaded or control), *organism* (plant, mammal, mussel), and their interactions.** In Model 1 we included *studyID* as random factor, while in Model 2 we included both *studyID* and *environment* (soil or water) as random factors.
(PDF)

**S5 Table. Comparison of the relative proportion of each bacterial family between control and invaded environments.** Differences are assessed using a linear mixed-effects model testing the normalized proportions of each bacterial family against *sample type* (Model 1 in S4 Table).
(PDF)

## Acknowledgments

We would like to thank Kali Mattingly (Ohio State University, USA) and Davide Rassati (Università degli Studi di Padova, Italy) for their helpful comments on the manuscript.

## Author Contributions

**Conceptualization:** Antonino Malacrinò, Victoria A. Sadowski, Tvisha K. Martin, Nathalia Cavichiolli de Oliveira, Ian J. Brackett, James D. Feller, Kristian J. Harris, Orlando Combita Heredia, Rosa Vescio, Alison E. Bennett.

**Data curation:** Antonino Malacrinò, Victoria A. Sadowski, Tvisha K. Martin, Nathalia Cavichiolli de Oliveira, Ian J. Brackett, James D. Feller, Kristian J. Harris, Orlando Combita Heredia, Rosa Vescio.

**Formal analysis:** Antonino Malacrinò, Victoria A. Sadowski, Tvisha K. Martin, Nathalia Cavichiolli de Oliveira, Ian J. Brackett, James D. Feller, Kristian J. Harris, Orlando Combita Heredia, Rosa Vescio.

**Methodology:** Antonino Malacrinò.

**Supervision:** Antonino Malacrinò, Alison E. Bennett.

**Writing – original draft:** Antonino Malacrinò.

**Writing – review & editing:** Antonino Malacrinò, Victoria A. Sadowski, Tvisha K. Martin, Nathalia Cavichiolli de Oliveira, Ian J. Brackett, James D. Feller, Kristian J. Harris, Orlando Combita Heredia, Rosa Vescio, Alison E. Bennett.

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
