## [Decision Letter · Decision Letter 0]

3 Aug 2020

PONE-D-20-18642

Biological invasions alter environmental microbiomes: a meta-analysis

PLOS ONE

Dear Dr. Malacrinò,

Thank you for submitting your manuscript to PLOS ONE. After careful consideration, we feel that it has merit but does not fully meet PLOS ONE’s publication criteria as it currently stands. Therefore, we invite you to submit a revised version of the manuscript that addresses the points raised during the review process.

Dear Author, as you can see, the reviewers raise many doubts on the soundness of your data and conclusions drawn from them. I expect you to submit a complete and thorough rebuttal to their comments and a revised version of your manuscript that will undergo an additional round of reviews.

We look forward to receiving your revised manuscript.

Kind regards,

Raffaella Casotti

Academic Editor

PLOS ONE

Journal Requirements:

2. Please include a caption for figure 1.

Additional Editor Comments (if provided):

Dear authora, as you will see, the reviewers raise many concern about the soundness of the data presented and on the suitability for publication. I suggest to submit a thorough rebuttal to their criticisms and to submit a revised version of your manuscript, which will undergo an additional round of revision.

Reviewers' comments:

Reviewer's Responses to Questions

**Comments to the Author**

1. Is the manuscript technically sound, and do the data support the conclusions?

Reviewer #1: Partly

Reviewer #2: No

Reviewer #3: Partly

2. Has the statistical analysis been performed appropriately and rigorously? 

Reviewer #1: Yes

Reviewer #2: No

Reviewer #3: Yes

3. Have the authors made all data underlying the findings in their manuscript fully available?

Reviewer #1: Yes

Reviewer #2: Yes

Reviewer #3: Yes

4. Is the manuscript presented in an intelligible fashion and written in standard English?

Reviewer #1: Yes

Reviewer #2: Yes

Reviewer #3: Yes

5. Review Comments to the Author

Reviewer #1: I believe this mansucript addresses an important question, and that the authors have done what they can with very limited numbers of studies available for analysis. I am left wondering if 5 published studies is an adequate sample number, particulalry considering that 2 are from very different systems to the other three, to draw broad conculions. The work seems very preliminary based on such a small sample size.

I have attached a PDF file with some minor comments.

Reviewer #2: This paper is not a meta-analysis. It is an analysis of combined data from five studies, but meta-analysis procedures were not used. Unfortunately, I cannot comment on the microbial data methods and analysis in this paper because it is beyond my area of expertise, but for reasons outlined below, I do feel the paper would be better framed as a review + additional analysis, rather than a meta-analysis (which it is not). And the conclusions need to be tempered and put in a greater context. The paper makes broad statements such as, “We show biological invasions decrease the diversity of environmental microbiomes,” which I do not feel are at all justified by the analyses. First, the study includes data from only five papers, three on plants (affecting soil), one on a mammal (affecting soil), and one on an aquatic snail (affecting water). This data set is far too small and disparate to use for sweeping generalizations. Second, it is not at all clear that the results support the statements. For example, a quick look at Figures 2A and 2C would indicate that there are absolutely no differences in either diversity or abundance between control and invaded sites, yet statistical analyses report differences. This makes me question the statistical models used. For example, the mixed model assessing diversity using environment as a random effect is inappropriate because you need more levels (at least 5) for an effect to be random. The unbalanced groups (3 plant, 1 mammal and 1 aquatic snail) could also cause model problems. At the very least, the model diagnostics need to be discussed to show that the model was appropriate. Finally, I think more needs to be known about individual studies to make any firm conclusions. For example, in study [28] natives were planted in polycultures of up to 7 species and non-natives were planted in monocultures. Does the difference in microbial diversity in such a study represent a realistic shift caused by an invader, or is it a reflection of the experimental design? I think more generalizations could be made if each of the five studies were discussed in some detail and the results put in first a specific context, and then if patters support it, into a more general context.

Reviewer #3: Dear editor,

This study is a meta-analysis of previously published metabarcoding data that compares environmental microbiomes in invaded and uninvaded sites in order to investigate the impacts of invasive species on microbiome diversity and community composition. Individual studies have had inconsistent results, and this is the first study that attempts to quantitatively draw broad patterns in this way. After filtering to control for quality and methodology, the authors were left with only five studies in their analysis. Their data showed an overall significant effect of invasive species on microbiomes: Shannon diversity decreased, and community composition shifted in taxa-specific ways at the family level.

Although this study’s methodology was well thought-out and executed rigorously, I am not sure that all of the authors’ claims are robustly supported. In the Discussion, the authors explain the limitations and caveats of this study very well. For example, the five studies are mostly about plants, and mostly concern soil microbiomes. Therefore, although I am not refuting the results themselves, I think that some of the conclusions that the authors draw are overly broad. In some places, their claims are well-qualified (line 23: “Our findings suggest that non-native species are responsible…”; line 232: “Without sufficient diversity of sample environments, it is impossible to tell…”), but others seem to overreach. The first line of the Discussion (line 117: “We show biological invasions decrease the diversity of environmental microbiomes.”) feels too broad to me. Although the results did indeed show a decrease in diversity, I think this sentence would benefit from a clause acknowledging that the environmental sampling is limited and biased. The rest of the Discussion does exactly this—pointing out all the ways this sample is biased (soils, invasive plants, northern hemisphere, summer)—so it seems odd to me that this claim is so broad. Alongside these caveats, the Discussion also excellently highlights the overall knowledge gap revealed by this meta-analysis.

I also thought that the discussion of taxa-level shifts was very interesting. Connecting microbial family shifts to possible shifts in community function is intriguing, although I am very glad that the authors acknowledge that amplicon sequencing cannot definitively show changes in function. In this section, as in the rest of the Discussion, the authors combine the results of their meta-analysis with results of previously published studies smoothly and effectively.

Overall, this is an interesting manuscript with a compelling question. However, given the dearth of compatible studies in the final meta-analysis, I think that this manuscript raises many more questions than it answers—this manuscript’s strength is in its ‘call to arms’ for more research rather than in the results themselves.

6. PLOS authors have the option to publish the peer review history of their article (what does this mean?). If published, this will include your full peer review and any attached files.

Reviewer #1: No

Reviewer #2: No

Reviewer #3: No

---

## [Author Response · Author response to Decision Letter 0]

24 Aug 2020

Response to reviewers’ comments (MS PONE-D-20-18642– REV1)

Answers to reviewers are reported after each comment

Reviewer #1

Comment #1 - I believe this manuscript addresses an important question, and that the authors have done what they can with very limited numbers of studies available for analysis. I am left wondering if 5 published studies is an adequate sample number, particularly considering that 2 are from very different systems to the other three, to draw broad conclusions. The work seems very preliminary based on such a small sample size.

Answer: Thank you! Yes, we agree that our analysis would benefit from a higher number of studies. In principle, a meta-analysis is defined as “the statistical combination of results from two or more separate studies” . Also, looking at the published literature (references 37-49 in our study), previous meta-analyses focused on a wide range of studies spanning from 2 (Bisanz et al.) to 30 (Duvallet et al.), with several of them in the range between 2 and 9 (Bisanz et al, Walters et al, Krych et al., Jiao et al., Wirbel et al., Rocca et al.). In addition, we took several steps to guarantee that data analysis would consider the heterogeneity between studies into account. Therefore, we believe that the number of studies and the approach we took guarantee a low bias in our results. We acknowledge these limitations in our discussions (L199-201, clean copy).

Comment #2 - I have attached a PDF file with some minor comments.

Answer: We integrated your suggestions in our manuscript, including a brief description of violin plots in the caption of Fig. 1A (L184-185, clean copy).

Reviewer #2

Comment #1 - This paper is not a meta-analysis. It is an analysis of combined data from five studies, but meta-analysis procedures were not used. Unfortunately, I cannot comment on the microbial data methods and analysis in this paper because it is beyond my area of expertise, but for reasons outlined below, I do feel the paper would be better framed as a review + additional analysis, rather than a meta-analysis (which it is not). 

Answer: We thank the reviewer for this comment; however, we respectfully disagree with their point of view. A meta-analysis is defined as “the statistical combination of results from two or more separate studies”1. The NIH NCI defines the meta-analysis as “A process that analyzes data from different studies done about the same subject …” . Most of the published literature that used our approach define their analysis as “meta-analysis” (refs 37-49 in our paper). We acknowledge that using data for single points , rather than aggregated data, is quite unusual. Literature is rich of examples where a meta-analytical approach has been used on individual “samples” . Therefore, we believe that meta-analysis is the most appropriate term to use in our case.

Comment #2 - And the conclusions need to be tempered and put in a greater context. 

Answer: Thanks for this suggestion. We added a sentence to the last paragraph to put our results in a greater context (L259-260, clean copy).

Comment #3 - The paper makes broad statements such as, “We show biological invasions decrease the diversity of environmental microbiomes,” which I do not feel are at all justified by the analyses. 

Answer: We included a statement (L199-201, clean copy) where we acknowledge the limits of our analysis.

Comment #4 - First, the study includes data from only five papers, three on plants (affecting soil), one on a mammal (affecting soil), and one on an aquatic snail (affecting water). This data set is far too small and disparate to use for sweeping generalizations. 

Answer: As we mentioned in our answer to comment #1, a meta-analysis can be done with even 2 studies. Furthermore, our analysis considers a total of 335 samples and we took a series of steps (L111-124, clean copy) to alleviate potential sources of bias. We acknowledge the limitations of our study throughout the discussions.

Comment #5 - Second, it is not at all clear that the results support the statements. For example, a quick look at Figures 2A and 2C would indicate that there are absolutely no differences in either diversity or abundance between control and invaded sites, yet statistical analyses report differences. This makes me question the statistical models used. For example, the mixed model assessing diversity using environment as a random effect is inappropriate because you need more levels (at least 5) for an effect to be random. The unbalanced groups (3 plant, 1 mammal and 1 aquatic snail) could also cause model problems. At the very least, the model diagnostics need to be discussed to show that the model was appropriate. 

Answer: Thanks for this suggestion. We ran all our models including or excluding “environment” as random effect, and we reported the comparison in the supplementary material. Overall, we found no differences between the two modelling strategies.

Comment #6 - Finally, I think more needs to be known about individual studies to make any firm conclusions. For example, in study [28] natives were planted in polycultures of up to 7 species and non-natives were planted in monocultures. Does the difference in microbial diversity in such a study represent a realistic shift caused by an invader, or is it a reflection of the experimental design? I think more generalizations could be made if each of the five studies were discussed in some detail and the results put in first a specific context, and then if patters support it, into a more general context.

Answer: Thanks for this suggestion. We agree with this comment and we provided a description of the individual studies (L94-106, clean copy). Discussion of our results in the context of individual studies was already provided (L196-201 and L205-215, clean copy).

Reviewer #3

Comment #1 - In the Discussion, the authors explain the limitations and caveats of this study very well. For example, the five studies are mostly about plants, and mostly concern soil microbiomes. Therefore, although I am not refuting the results themselves, I think that some of the conclusions that the authors draw are overly broad. In some places, their claims are well-qualified (line 23: “Our findings suggest that non-native species are responsible…”; line 232: “Without sufficient diversity of sample environments, it is impossible to tell…”), but others seem to overreach. The first line of the Discussion (line 117: “We show biological invasions decrease the diversity of environmental microbiomes.”) feels too broad to me. Although the results did indeed show a decrease in diversity, I think this sentence would benefit from a clause acknowledging that the environmental sampling is limited and biased. The rest of the Discussion does exactly this—pointing out all the ways this sample is biased (soils, invasive plants, northern hemisphere, summer)—so it seems odd to me that this claim is so broad. Alongside these caveats, the Discussion also excellently highlights the overall knowledge gap revealed by this meta-analysis.

Answer: Thanks for your suggestion. We included it in the manuscript (L199-201, clean copy).

Comment #2 - I also thought that the discussion of taxa-level shifts was very interesting. Connecting microbial family shifts to possible shifts in community function is intriguing, although I am very glad that the authors acknowledge that amplicon sequencing cannot definitively show changes in function. In this section, as in the rest of the Discussion, the authors combine the results of their meta-analysis with results of previously published studies smoothly and effectively.

Answer: Thank you!

Comment #3 - Overall, this is an interesting manuscript with a compelling question. However, given the dearth of compatible studies in the final meta-analysis, I think that this manuscript raises many more questions than it answers—this manuscript’s strength is in its ‘call to arms’ for more research rather than in the results themselves.

Answer: Thank you!

---

## [Decision Letter · Decision Letter 1]

21 Sep 2020

PONE-D-20-18642R1

Biological invasions alter environmental microbiomes: a meta-analysis

PLOS ONE

Dear Dr. Malacrinò,

Thank you for submitting your manuscript to PLOS ONE. After careful consideration, we feel that it has merit but does not fully meet PLOS ONE’s publication criteria as it currently stands. Therefore, we invite you to submit a revised version of the manuscript that addresses the points raised during the review process.

We look forward to receiving your revised manuscript.

Kind regards,

Raffaella Casotti

Academic Editor

PLOS ONE

Additional Editor Comments (if provided):

Dear authors, the reviewers recognize the value of your work but raise some additional concern about conclusions drawn from statistical tests. I kindly ask you to address the point raised by acknowldging it in the etxt and commenting further the limitations of the analyses and of the limited dataset analyzed. Please, do NOT simply rebutt the comment but acknowledge it and comment it appropriately supporting your statement with solid arguments.

Thanks for submitting you best work to this journal

Reviewers' comments:

Reviewer's Responses to Questions

**Comments to the Author**

1. If the authors have adequately addressed your comments raised in a previous round of review and you feel that this manuscript is now acceptable for publication, you may indicate that here to bypass the “Comments to the Author” section, enter your conflict of interest statement in the “Confidential to Editor” section, and submit your "Accept" recommendation.

Reviewer #1: (No Response)

2. Is the manuscript technically sound, and do the data support the conclusions?

Reviewer #1: (No Response)

3. Has the statistical analysis been performed appropriately and rigorously? 

Reviewer #1: I Don't Know

4. Have the authors made all data underlying the findings in their manuscript fully available?

Reviewer #1: Yes

5. Is the manuscript presented in an intelligible fashion and written in standard English?

Reviewer #1: Yes

6. Review Comments to the Author

Reviewer #1: The authors have addressed the concerns I had with the paper to what appears to be a reasonable extent. They argue that their data set are adequate to address their questions, and provide an explanation why they believe this is the case. I find this hard to judge, and I expect the other reviewers will comment further on the matter. My feeling that the work is quite preliminary stands, but I am happy to be convinced otherwise.

I was concerned with the response to reviewer #2, who commented on the statistical analyses. I do agree that many of the comparisons declared statistically different certainly do not appear to be, from the data shown in the figure. E.g. Fig 1c, it seems very unlikely that Sphingomonadaceae were different in control vs invaded – the medians are essentially identical; yet Gemmatimonadaceae are not different? There are a number of such instances in the figures, and the authors need to explain this clearly. I suspect the asterisks indicating significance are incorrectly annotated, maybe? Their assertion that including as a random effect in their analysis and finding it made no difference to the conclusions, does not go a long way to explaining the apparent lack of differences shown in the figures, despite the pairs being labelled as significantly different. Fig 1a likewise seems to show two very similar data distributions, with the medians essentially the same.

7. PLOS authors have the option to publish the peer review history of their article (what does this mean?). If published, this will include your full peer review and any attached files.

Reviewer #1: No

---

## [Author Response · Author response to Decision Letter 1]

1 Oct 2020

Reviewer #1

Comment #1 - I was concerned with the response to reviewer #2, who commented on the statistical analyses. I do agree that many of the comparisons declared statistically different certainly do not appear to be, from the data shown in the figure. E.g. Fig 1c, it seems very unlikely that Sphingomonadaceae were different in control vs invaded – the medians are essentially identical; yet Gemmatimonadaceae are not different? There are a number of such instances in the figures, and the authors need to explain this clearly. I suspect the asterisks indicating significance are incorrectly annotated, maybe? Their assertion that including as a random effect in their analysis and finding it made no difference to the conclusions, does not go a long way to explaining the apparent lack of differences shown in the figures, despite the pairs being labelled as significantly different. Fig 1a likewise seems to show two very similar data distributions, with the medians essentially the same.

Answer: We agree with the reviewer that our graphs do not properly show the differences we claim in our text. Thus, we changed our Fig 1: panel A has been moved to the text (L167-168), panel B is now the main figure, and panel C has been moved to S5 table. Furthermore, we integrated our discussion to reflect your comments (L200-203 and L217-218).

---

## [Editor Report · Decision Letter 2]

7 Oct 2020

Biological invasions alter environmental microbiomes: a meta-analysis

PONE-D-20-18642R2

Dear Dr. Malacrinò,

We’re pleased to inform you that your manuscript has been judged scientifically suitable for publication and will be formally accepted for publication once it meets all outstanding technical requirements.

Kind regards,

Raffaella Casotti

Academic Editor

PLOS ONE

Additional Editor Comments (optional):

Dear authors, I think that your manuscript is now suitable for publication
---

## [Editor Report · Acceptance letter]

14 Oct 2020

PONE-D-20-18642R2 

Biological invasions alter environmental microbiomes: a meta-analysis 

Dear Dr. Malacrinò:

I'm pleased to inform you that your manuscript has been deemed suitable for publication in PLOS ONE. Congratulations! Your manuscript is now with our production department. 

Kind regards, 

on behalf of

Dr. Raffaella Casotti 

Academic Editor

PLOS ONE